# Differences in Neuropsychological Performance between Refugee and Non-Refugee Children in Palestine

**DOI:** 10.3390/ijerph18115750

**Published:** 2021-05-27

**Authors:** Ahmed F. Fasfous, María Nieves Pérez-Marfil, Francisco Cruz-Quintana, Miguel Pérez-García, Hala R. Al-Yamani, Manuel Fernández-Alcántara

**Affiliations:** 1Social Sciences Department, Bethlehem University, Bethlehem 92248, Palestine; afasfous@bethlehem.edu; 2Mind, Brain and Behavior Research Center (CIMCYC), University of Granada, 18010 Granada, Spain; fcruz@ugr.es (F.C.-Q.); mperezg@ugr.es (M.P.-G.); 3Faculty of Psychology, University of Granada, 18010 Granada, Spain; 4Faculty of Education, Bethlehem University, Bethlehem 92248, Palestine; halay@bethlehem.edu; 5Department of Health Psychology, University of Alicante, 03690 Alicante, Spain; mfernandeza@ua.es

**Keywords:** refugee, neurodevelopment, cognitive, neuropsychological assessment

## Abstract

Neuropsychological studies on refugee children are scarce, but there are even less in the case of Palestinian children. This work aims to study the neuropsychological performance of Palestinian refugee children in Palestine compared to other Palestinian children living outside refugee camps. A comprehensive Neuropsychological battery was administrated to 584 Palestinian school children (464 refugees and 120 non-refugees) aged 6, 7, and 8 years old. Results showed that non-refugee children outperformed refugee children in sustained attention, verbal comprehension, verbal memory, and visual memory. This study is the first to have performed a comprehensive neuropsychological assessment, based on a standardized and validated battery with the Palestinian refugee children. It supports professionals in their evaluation of neurodevelopment and neuropsychological alterations in refugee and non-refugee children in Palestine.

## 1. Introduction

According to the data on worldwide refugees included in the ‘Global Trends’ report [1], by the end of 2018, there were 70.8 million forcibly displaced persons worldwide: 41.3 million internal displacements, 25.9 million refugees, and 3.5 million asylum seekers. All were forced to flee due to persecution, conflict, violence, or human rights violations.

Palestinian refugees stand out among these populations, for the duration of the situation and their sheer number: they account for a quarter of the world’s refugees. June 2020 marked the 72nd year of the Israeli occupation of Palestine. Over 5.3 million Palestinian refugees (living in the Gaza Strip, the West Bank, Syria, Lebanon, and Jordan) undergo extreme restrictions of freedom of movement, affecting their access to school, employment, agricultural land, medical services, and family relationships [2]. The Gaza Strip has a population of more than 1.9 million people, of which 1.4 million are refugees. Some 2.4 million live on the West Bank area and a quarter of refugees live in 19 camps run by the United Nations Relief and Works Agency (UNRWA). The UNRWA currently provides basic health, education, and health infrastructure services to these refugee camps [3].

Palestinian refugee camps face several problems, notably: high population density, high unemployment, poverty, lack of space, parks and play areas, shortages of health and education services, as well as poor infrastructure. Children aged under 14 years make up nearly 40% of the Hebron and Bethlehem camps, and schools in these camps are immensely overcrowded [3].

The evidence of the psychopathological effects of war on children and adolescents is extensive. Various studies have analyzed the impact of exposure to war conflict on the mental health of children and adolescents [4,5,6,7,8]. It has been found that these children experience intense acute stress and many develop Post-Traumatic Stress Disorder (PTSD) [9,10]. Common symptoms and reactions after a traumatic event include sadness, anger, fears, numbness, nervousness, moodiness or irritability, changes in appetite, difficulty sleeping, nightmares, avoidance of situations reminiscent of the trauma and deteriorating concentration [11].

Research in Palestine has shown that as a result of the war, children living in the West Bank, Hebron, and Gaza areas often have mental health problems [10,12,13,14,15] especially PTSD, as well as anxiety and depression disorders [13,16]. It is estimated that approximately 41% of children living in Gaza experience post-traumatic stress and depression symptoms [10,12,15] and 77.4% of children have shown moderate or severe PTSD symptoms in the Hebron area, of which 20.5% meet the diagnostic criteria for PTSD [17]. The number of traumatic experiences related to conflict correlates positively with the prevalence of mental, behavioral and emotional problems. The main determining factors identified are the level and type of exposure, age, gender, socioeconomic adversity, social support, and religiosity [18]. In this line, the study conducted by Thabet and colleagues [16] in the Gaza Strip found high percentages of psychopathology in that population. A total of 59%presented a wide range of clinical PTSD symptoms, 21.9% of children presented anxiety, and 50.6% had depression. In addition, the total number of traumatic events was significantly related to the presence of PTSD, avoidance behaviors, physiological activation, anxiety, and depression [9]. Results showed that political violence due to war trauma was related to the development of PTSD and depression in Palestinian children in the Gaza Strip [16].

Although children and adolescents account for a big share of the refugee population, the neuropsychological problems of children and young people have not been sufficiently addressed in the scientific literature, unlike studies on psychopathology [19,20]. It has been suggested that traumatic experiences related to war conflicts not only increase the risk of mental health problems but may also have consequences on neurocognitive and socio-emotional development [7,8,21]. The findings suggest that war-related trauma has an impact on cognitive-affective processes that are essential to the healthy development of both children and adolescents [22]. In this way, toxic stress has been found to affect the development of certain brain structures. It occurs when the child is exposed to constant and uncontrollable unfavorable experiences that he is unable to adequately cope with [23] Toxic stress has been shown to cause a reduction in the size of areas such as the corpus callosum, hippocampus, amygdala and to attenuate the development of the left neocortex.

As indicated above, neuropsychological studies on children exposed to conflict are scarce, but there are even less in the case of Palestinian children. In a study by Qouta et al., [14] on Palestinian children exposed to traumatic experiences, the authors found that these children also suffered from concentration, attention, and memory problems. In a more recent study, Buckner et al., [24] found that conflict exposure could negatively influence the development of executive functions, which can in turn affect academic performance. Their analysis focused on a sample of 185 Young Palestinians, aged between 6 and 16 years. They concluded that students from schools that were the most exposed to political conflict achieved lower levels of executive functioning, notably planning. In short, the few studies that have been conducted indicate that Palestinian children present neuropsychological alterations.

Nevertheless, further studies are necessary to replicate these results. They should be conducted using extensive neuropsychological testing batteries to obtain a complete profile of all neuropsychological domains. Furthermore, these neuropsychological tests should be adapted to the Arab population, despite a shortage of such tests for this population [25]. In addition, studies should consider the degree of exposure to conflict [26], because the socio-economic situation and exposure to violence in Palestine may vary depending on whether children live inside or outside a refugee camp. Finally, the studies should verify whether the levels of exposure to conflict are even across different years of age or whether they shift depending on the children’s age.

Based on all the above, the objective of the present work was to compare the neuropsychological functioning of Palestinian children living in refugee camps (who are exposed to violence the most) compared to that of Palestinian children living outside the camps (who are less exposed to violence), using a standardized and validated battery on the Arab population (Child Computerized Neuropsychological Assessment Battery, BENCI) [27] which evaluates the major neuropsychological domains. In addition, to check whether exposure to violence has equal effects according to age and sex, the study was conducted on children aged 6, 7 and 8 years. These age groups were selected because they correspond to the first years of schooling in the Palestinian education system and it is at these ages that the neuropsychological problems relating to school performance can appear. In addition, several studies have stressed the importance of conducting neuropsychological studies with schoolchildren and the need to create normative data on school children. We hypothesized that Palestinian children exposed to violence (i.e., living in refugee camps) would present more extensive alterations regarding memory, attention, and executive function [14,24]. No hypotheses were advanced in any of the other domains because of the absence of any previous studies.

## 2. Materials and Methods

### 2.1. Participants

A total of 584 Palestinian schoolchildren (295 boys, 50.51%; 289 girls, 49.49%) aged 6, 7, and 8 voluntarily participated in the study (mean (M) = 6.99, standard deviation (SD) = 0.817). As noted above, these ages were selected because they correspond to the first years of schooling in the Palestinian education system and due to the scarcity of standard scale tests for these ages. Participants were selected from two Palestinian provinces (Hebron and Bethlehem). These two geographical areas include five refugee camps, in Bethlehem: the Dheisheh Camp (established in 1949), the Beit Jibrin Camp (established in 1950), the Aida Camp (established in 1950), and in Hebron: the Arroub Camp (established in 1949), and the Fawwar Camp (established in 1949). A total of 464 (79.45%) children were selected from schools located within five refugee camps. In addition, 120 (20.55%) children living outside refugee camps participated in the study. Table 1 presents the geographical distribution of the study sample.

Table 2 includes general information on the sample, as well as sociodemographic data on the parents.

### 2.2. Instruments

The computerized Battery for the Neuropsychological Evaluation of Children (BENCI) consists of 14 neuropsychological tests that measure different neuropsychological functions. They allow a comprehensive evaluation of children’s neurodevelopment. The main functions are: language (Verbal Comprehension and Phonemic Fluency), Sustained Attention (Continuous Performance Test), Visuomotor Coordination (Visuomotor test), Verbal Memory (Immediate Recall, Delayed Recall, and Recognition), Visual Memory (Immediate Recall, Delayed Recall, and Recognition), and Executive Function (Working Memory, Planning, Abstract Reasoning, Semantic Fluency, and Alternate Visuomotor Test). Further details on the tests used to measure these functions can be found in previous studies [27,28,29].

BENCI has been used in different contexts to evaluate neurodevelopment and has shown good psychometric properties [30]. In addition, this battery has been validated and adapted to Arab culture and its good reliability and validity on Arab children has been demonstrated [27].

### 2.3. Procedure

First, the study was approved by the ethical commissions of the University of Granada and Bethlehem University as a cooperative project between the two universities on 6 October 2014 (Reference No. 201402400001070). Once the authorization was obtained from the ethics committees, an authorization from the “Directorate of Education, Bethlehem” was obtained to conduct the study at public schools outside refugee camps. Another authorization was obtained from the office of The United Nations Relief and Works Agency for Palestine Refugees (UNURWA) in Palestine to conduct the study at schools located in Hebron and Bethlehem refugee camps. The schools were randomly selected and, applying tiered sampling to the class lists, the sample was subdivided into three levels (1st, 2nd and 3rd year) based on gender and academic level. The children were then randomly extracted from each strata. The 1st year corresponds to children aged 6 years, the 2nd year to children aged 7 years, and the 3rd grade to children aged 8 years.

All assessments were conducted in the morning, during school hours, in a room located within the school premises. In this study, the order of administration was the same for all participants, with amid-session 10-min break. Administering the battery lasted on average 75 min and the order of administration was determined following the recommendations of Lezak et al. [31].

### 2.4. Statistical Analysis

To start with, descriptive analyses were conducted using the main socio-demographic variables. ANOVAs were then conducted with a 2 × 2 × 3 inter subject factorial design, considering the group (living inside the refugee camp vs. outside the refugee camp), sex (male vs. female) and age (six vs. seven vs. eight years of age) as independent variables, and BENCI subtest results as dependent variables. In the case of the verbal memory subtest, an extra within factor was added to the design (immediate recall trials 1, 2, 3, delayed trials). In a similar way, an extra within factor was added for visual memory (immediate and delayed trials) as well as a go/no-go task (trial blocks 1–4). Some post hoc Bonferroni analyses were conducted when significant statistical interactions were found. Furthermore, partial eta squared was calculated to measure the effect size. The significant levels were adjusted to 0.05 and all analyses were conducted using SPSS 23.

## 3. Results

### 3.1. Language

Main effect of age was found to have an impact across all the tests used to evaluate language. Post hoc comparisons indicated that children aged 8 years performed better than children aged 7 years, and children aged 7 years performed better than children aged 6 years in the phonetic fluency test. However, no statistically significant differences were found in the verbal comprehension tests (images and figures) between children aged 7 and 8 years.

In the image comprehension test, we found main effect of area and the scores of the refugee children were lower than that of children outside the camps. In addition, another main effect was found for sex with girls performed generally better than boys in the phonetic fluency test. No interactions were found for any language subtest (see Table 3).

### 3.2. Sustained Attention

The Continuous Performance Tasks (CPT) results main effect for correct answer and omissions, showing that the group of refugee children was more successful than children outside the camp. However, non-refugee children made fewer mistakes than refugee children. Considering the time variable, the results showed that refugee children were slower at completing the test. Regarding main effect of sex, boys were faster at responding to stimuli. The results demonstrated a Sex × Area interaction, refugee boys and girls showing similar values, although in the case of non-refugee children, girls performed significantly better than boys in this test (see Table 3).

### 3.3. Visuomotor

In the visuomotor coordination test, we found main effect of age. Younger children took longer than older children to perform the task. The group of children aged 6 years scored lower than the two other groups. No statistically significant differences were found between the 7 and 8 year age groups. No interactions were found for visuomotor coordination (see Table 3).

### 3.4. Memory

For the recognition trials, the main effects of age were found for verbal memory and visual memory with younger children scoring lower than older ones (Table 3).

For the free recall trails and learning curve, we found the main effects of area and age in verbal and visual memory. Palestinian children living outside refugee camps performed better than refugee children in verbal memory tasks. In addition, the results presented notable differences between children based on age. Post hoc tests showed significant differences between the three age groups.

As for the verbal memory results, refugee children scored lower on visual memory tasks. Significant differences were found; however, between the children aged 6 and 7 years, the older group scored higher. They were also found between children aged 6 years and the group of children aged 8 years, with the older group scoring higher. These differences were significant between the group of children aged 7 and the group of children aged 8. No interactions were found for memory (see Table 4).

### 3.5. Executive Functions

The results revealed significant differences between scores in the case of 4 out of 5 executive function tests based on age (main effect). Specifically, the group of children aged 8 obtained higher scores than the group of children aged 6 and 7 year. The group of children aged 7 years scored higher than the group of children aged 6 years on working memory, abstract reasoning, and semantic fluency tasks (Table 3).

In relation to the Alternate Visuomotor task, the results showed significant differences between the three age groups and sex (main effect of age and sex). Post hoc comparisons indicated that the older groups achieved higher scores. Moreover, girls outperformed boys in this test. No interactions were found for executive functions (see Table 4).

In addition, our results showed that effect size was large in sustained attention (Area), while medium effect sizes were reported in Phonemic fluency (sex), visuomotor coordination (sex), Alternate Visuomotor (sex), Working Memory (sex), abstract reasoning (sex), and verbal memory recognition (age). Furthermore, results revealed a small effect size in the rest of the neuropsychological tests (see Table 5).

## 4. Discussion

The main objective of the present study was to assess the neuropsychological performance of refugee children in Palestine compared to other Palestinian children living outside refugee camps. Though several papers have examined the mental health of Palestinian children, few studies have studied neuropsychological performance [14]. This study is the first to have used a complete neuropsychological battery, including major neuropsychological domains, to assess the neuropsychological performance of Palestinian refugee and non-refugee children. Moreover, it is also the first study to have examined the impact of living in a refugee camp on neuropsychological performance during early school years.

Overall, the results showed that refugee children perform worse than non-refugee children in sustained attention, verbal comprehension, verbal memory, and visual memory. This finding is consistent with that of other prior studies that concluded that refugees tend to perform worse neuropsychologically. A recent study showed the deterioration of sustained attention during an inhibitory control task (Stroop’s task) in young people who had experienced the loss of their parents and who lived in a conflict zone [32].

These results can be explained by the socio-economic difficulties of refugee children. In addition to the marginalization and difficult living conditions within refugee camps [3], Palestinian camps are the targets of on-going incursions by Israeli occupying forces. Therefore, the probability of children experiencing traumatic events is higher, and this could explain the poor performance of children who live in refugee camps. Neuropsychological functions such as attention and memory are among the functions that are the most affected by exposure to psychological trauma [22,33,34]. In the same line, young Afghan refugees have been found to present poorer working memory capacity than a control group [6].

The absence of differences in other neuropsychological functions could be related to the similar context of Palestinian children living outside and inside refugee camps. Both Palestinian children living inside and outside refugee camps are exposed to various forms of political, economic, and social violence, as they all live under Israeli occupation control. However, children living within refugee camps are subject to worse life conditions and are also exposed to more extensive violence. These similarities and differences in the contexts in which children grow could explain the child performance similarities and differences in neuropsychological tests.

With respect to gender differences in neuropsychological performance among Palestinian children, three main functions were identified: phonetic fluency, sustained attention, visuomotor coordination. These findings coincide with that of other studies in which girls presented a better performance in language tests [35,36], visuomotor coordination [37,38] and sustained attention [39,40]. However, there is no scientific consensus regarding these differences. In our study, these differences were encountered in the case of some functions, but we must remember that both boys and girls were exposed to a general context of violence. Added to the above differences, non-refugee Palestinian girls performed better than the group of non-refugee boys in the visual memory test. However, these differences were not detected between the boys and girls in the refugee camps. The results can be explained by the higher level of stress children are subjected to when living in refugee camps. As indicated, high levels of exposure to violence experienced in refugee camps may have reduced gender differences.

The age variable had a clear impact on children’s neuropsychological performance in all BENCI tests except the sustained attention test. These results are similar to that of prior studies that used the BENCI battery to measure neurodevelopment in children in Morocco [17] and Ecuador [28]. Improvements in the performance of neuropsychological tests reflect the normal development of neuropsychological functions during childhood [41]. None of the analyses, however, showed an age interaction per group, indicating that children living inside and outside refugee camps had improved their neuropsychological performance at different ages in a similar way. Since it is not possible to study a group of Palestinian children that is “not exposed” to violence, we are unable to verify whether such patterns of performance increase with age follow an adequate or altered pattern.

The main risk factors for normal neurocognitive development are related to conditions of the prenatal stage, perinatal circumstances, nutritional, infectious and toxic factors, and child-rearing practices [42], as well as socioeconomic conditions, in especially those related to poverty, in general, and extreme poverty in particular [43] and with exposure to violence [23]. Many of these circumstances are present in the sample of this study. This work constitutes the first neuropsychological study carried out with children living in Palestinian refugee camps. It has made it possible to obtain a baseline that can be used for future longitudinal studies. The results confirm the need to generate programs to intervene in the detected deficits and implement other preventive programs from an earlier age. These results have important health implications, indicating the vulnerability and the negative effects in the neurodevelopment in these children. Given the large number of variables and factors that interact (poverty, lack of space, poor infrastructure, and education services) it is necessary to develop interventions for health promotion that can address both internal and external risks factors.

The main limitations of the study include, first of all, the fact of having included only three age groups from two provinces of Palestine. Future studies should be performed, using samples that span a broader age range to thus obtain information on neuropsychological performance in adolescent populations in these contexts. Second, given Palestine’s socio-political situation of conflict, it was not possible to include a control group of Palestinian children that was “not exposed” to violence. Future studies might consider including a control group of Palestinian children living outside Palestine, such as Jordan, who are not exposed to conflict violence. Third, we could not assess the intellectual coefficient of the children that participated in the research. Finally, the cross-sectional research design does not allow us to study the direct influence of age in the neuropsychological development of these populations. Studies using a longitudinal design are needed to test the effects of trauma in the cognitive development of refugees and non-refugees children in Palestine

Despite this, the present work is the first to have performed a comprehensive evaluation of the neuropsychological functioning, based on a standardized and validated battery with the Arab population and during the first years of schooling. This battery can be used in future research to examine further variables and to create normalized data on Palestinian children. It also supports professionals in their evaluation of neurodevelopment and neuropsychological alterations in refugee and non-refugee Palestinian children in Palestine. Future studies should also consider comparing these results with that of Palestinian children living outside Palestine to better understand the role of continuous exposure to violence.

## 5. Conclusions

To conclude, data from our study indicate that Palestinian children living in refugee camps present lower neuropsychological test performance than other children in sustained attention, verbal comprehension, verbal memory, and visual memory.

## Figures and Tables

**Table 1 ijerph-18-05750-t001:** Geographical distribution of the study sample.

Province	Boys/Girls	Refugees	Non-Refugees	6 Years	7 Years	8 Years
Hebron	114/117	231	0	73	78	80
Bethlehem	181/172	233	120	123	117	113
Total	295/289	464	120	196	195	194

**Table 2 ijerph-18-05750-t002:** General information on the sample and sociodemographic data of the parents.

	Refugee M (SD) or *n* (%)	Non-Refugee M (SD) or *n* (%)	*p*
Number of people they live with	5.77 (1.91)	5.57 (1.91)	0.295
Number sisters and brothers	3.9 (2.02)	4 (2.2)	0.297
Hours of sleep	9.49 (1.27)	10.48 (1.94)	0.000
Chronic health problem Yes No	137 (29.6%) 326 (70.4%)	18 (15%) 102 (85%)	0.001
Father’s educational level No studies Primary High school Graduate University Master/Doctorate Do not know	30 (6.5%) 114 (24.6%) 197 (42.6%) 33 (7.1%) 70 (15.1%) 20 (4.1%)	5 (4.2%) 59 (49.2%) 43 (35.8%) 4 (3.3%) 8 (6.7%) 1 (0.8%)	0.000
Mother’s educational level No studies Primary High school Graduate University Master/Doctorate Do not know	12 (2.6%) 86 (18.5%) 205 (44.2%) 25 (5.4%) 124 (26.7%) 12 (2.6%)	4 (3.3%) 20 (16.7%) 59 (49.2%) 7 (5.8%) 29 (24.2%) 1 (0.8%)	0.778
Employeed father Yes No	402 (86.7%) 62 (13.3%)	114 (95%) 6 (5%)	0.043
Father’s job Autonomous Family business Unskilled manual labor Skilled manual labor Domestic/caregiver Professional Student	35 (8.7%) 2 (0.5%) 183 (45.5%) 37 (9.2%) 0 100 (24.9%) 45 (11.2%)	14 (12.3) 2 (1.7%) 70 (61.7%) 13 (11.4%) 0 10 (8.7%) 5 (4.2%)	0.000
Employeed motther Yes No	78 (16.8%) 386 (83.2%)	10 (8.3%) 110 (91.7%)	0.021
Mother’s job Autonomous Family business Unskilled manual labor Skilled manual labor Domestic/caregiver Professional Student	7 (9%) 0 5 (6.4%) 3 (3.8%) 3 (3.8%) 27 (34.6%) 33 (42.3%)	0 0 0 0 0 4 (40%) 6 (60%)	0.808

M = mean; SD = standard deviation.

**Table 3 ijerph-18-05750-t003:** Neuropsychological performance of Palestinian children.

Domain	Test	Refugee *n* = 464 M (SD)	Non-Refugee *n* = 120 M (SD)	Male *n* = 295 M (SD)	Female *n* = 289 M (SD)	6 *n* = 196 M (SD)	7 *n* = 195 M (SD)	8 *n* = 193 M (SD)	Main Effects and Interactions	*p*
Language	Phonemic Fluency	2.76 (2.16)	2.95 (2.13)	2.37 (1.94)	3.23 (2.27)	2.02 (1.66)	2.68 (1.96)	3.70 (2.42)	Sex (Female > male) Age (1 < 2 < 3)	0.000 0.000
	ComFig	8.52 (1.29)	8.33 (1.39)	8.39 (1.38)	8.57 (1.24)	8.03 (1.39)	8.58 (1.33)	8.84 (1.07)	Age (1 < 2; 1 < 3)	0
	ComIma	8.67 (1.25)	9.01 (1.07)	8.65 (1.26)	8.83 (1.17)	8.36 (1.34)	8.84 (1.69)	9.02 (1.05)	Ref < noR Age (1 < 2; 1 < 3)	0.003 0.000
Sustained Attention	CPT/CA	278.97 (25.60)	228.16 (101.53)	269.70 (54.29)	267.34 (56.18)	265.47 (55.12)	271.88 (51.66)	268.25 (58.71)	noR < Ref	0
	CPT/OM	10.58 (9.12)	5.84 (5.66)	9.26 (8.38)	9.96 (9.09)	11.24 (9.51)	8.94 (8.43)	8.61 (7.99)	noR < Ref Age (3 < 1)	0.000 0.030
	CPT/RT	0.63 (0.14)	0.35 (0.43)	0.59 (0.24)	0.56 (0.28)	0.57 (0.87)	0.57 (0.26)	0.58 (0.24)	Area (Ref > noR) Sex (Male > Female) Area × Sex	0.000 0.006 0.013
Visuomotor	Vis-motor 1	109.74 (55.02)	105.63 (51.20)	112.87 (60.14)	104.84 (47.23)	134.61 (66.01)	102.77 (45.88)	88.98 (35.63)	Age (1 > 2; 1 > 3)	0
Memory—Recognition	Verbal Memory	15.83 (2.45)	16.18 (2.11)	15.61 (2.63)	16.20 (2.07)	14.97 (2.68)	16.13 (2.28)	16.61 (1.81)	Age (1 < 2; 1 < 3)	0
	Visual Memory	41.73 (6.39)	42.70 (4.89)	41.05 (6.61)	42.83 (5.43)	40.17 (6.64)	42.17 (6.22)	43.47 (5.14)	Age (1 < 3) Area × Sex	0.001 0.042
Executive Functions	Working Memory	4.04 (1.93)	3.75 (1.90)	3.90 (1.94)	4.07 (1.91)	3.23 (1.48)	3.90 (1.74)	4.82 (2.16)	Age (1 < 2 < 3)	0
	Planning	11.09 (1.57)	11.45 (2.93)	11.01(1.85)	11.32 (2.00)	11.23 (2.14)	10.97 (1.60)	11.28 (2.01)	NS	NS
	Abstract Reasoning	12.12 (5.33)	11.70 (5.03)	11.72 (5.26)	12.35 (5.27)	9.92 (4.33)	11.96 (5.45)	14.25 (5.07)	Age (1 < 2 < 3)	0
	Semantic Fluency	8.42 (3.43)	8.42 (3.14)	8.35 (3.43)	8.49 (3.31)	7.13 (3.02)	8.35 (3.15)	9.79 (3.40)	Age (1 < 2 < 3)	0
	Vis-motor 2	123.61 (55.22)	117.94 (50.39)	127.66 (59.73)	117.12 (47.57)	145.05 (54.23)	122.21 (57.39)	99.72 (39.83)	Sex (Male > Female) Age (1 > 2 > 3)	0.016 0.000

ComFig = Verbal Comprehension—Figures; ComIma = Verbal comprehension images; CPT = Continuous performance Test; CA = Correct Answers; OM = omission; RT = Reaction time; Vis-motor 1 = visuomotor coordination; Vis-motor 2 = Alternate Visuomotor; M = mean; SD = standard deviation; Sex = main effect of sex variable; Age = main effect of age variable; Area = main effect of area variable; Area × Sex: interaction of area and sex variables; Ref = refugee group; noR = non-refugee group; NS = non-significant.

**Table 4 ijerph-18-05750-t004:** Verbal and visual memory tests performance.

Domain	Refugee *n* = 463 M (S. D)	Non-Refugee *n* = 119 M (SD)	Male *n* = 295 M (SD)	Female *n* = 287 M (SD)	6 *n* = 195 M (SD)	7 *n* = 194 M (SD)	8 *n* = 193 M (SD)	Main Effects and Interactions	*p*
Verbal Memory 1	4.15 (1.68)	4.57 (1.75)	4.14 (1.73)	4.33 (1.66)	3.74 (1.62)	4.28 (1.76)	4.68 (1.59)	Area (noR > Ref) Age (1 < 2 < 3)	0.041 0.000
Verbal Memory 2	5.68 (1.69)	5.85 (1.76)	5.74 (1.64)	5.68 (1.80)	5.21 (1.77)	5.68 (1.70)	6.25 (1.49)		
Verbal Memory 3	6.48 (1.60)	6.39 (1.69)	6.47 (1.60)	6.45 (1.64)	5.82 (1.67)	6.59 (1.56)	6.99 (1.40)		
Verbal Memory 4	5.49 (1.74)	5.81 (1.84)	5.60 (1.81)	5.51 (1.71)	4.93 (1.62)	5.62 (1.69)	6.11 (1.77)		
Visual Memory A	6.48 (2.14)	6.95 (2.12)	6.50 (2.14)	6.66 (2.14)	5.80 (2.08)	6.76 (2.03)	7.18 (2.08)	Area (noR > Ref) Age (1 < 2, 1 < 3)	0.000 0.000
Visual Memory B	4.65 (1.92)	5.35 (2.22)	4.45 (2.03)	4.79 (2.00)	3.91 (1.99)	4.94 (1.99)	5.01 (1.89)		

1 = Immediate Recall first attempt; 2 = Immediate Recall second attempt; 3 = Immediate Recall third attempt; 4 = Delayed Recall; A = Immediate Recall; B = Delayed Recall; Ref = refugees; noR = non-refugees; M = mean; SD = standard deviation.

**Table 5 ijerph-18-05750-t005:** Effect size and interpretation.

Domain	Test	Variable	Eta Squared $\left(\eta_{\;p}^{2}\right)$	Interpretation
Language	Phonemic Fluency	Area Age Sex Interaction	0.004 0.044 0.069 0.001	Small Small Medium Small
	Verbal Comprehension—Figures	Area Age Sex Interaction	0.002 0.001 0.051 0.005	Small Small Small Small
	Verbal Comprehension Images	Area Age Sex Interaction	0.016 0.001 0.032 0.000	Small Small Small Small
Sustained Attention	Continuous performance Test (Correct answers)	Area Age Sex Interaction	0.144 0.003 0.002 0.003	Large Small Small Small
	Continuous performance Test (Omission)	Area Age Sex Interaction	0.052 0.000 0.012 0.002	Small Small Small Small
	Continuous performance Test (Reaction Time)	Area Age Sex Interaction	0.195 0.013 0.000 0.001	Large Small Small Small
Visuomotor	Visuomotor Coordination	Area Age Sex Interaction	0.003 0.004 0.088 0.000	Small Small Medium Small
Executive Functions	Working Memory	Area Age Sex Interaction	0.002 0.003 0.087 0.001	Small Small Medium Small
	Planning	Area Age Sex Interaction	0.006 0.003 0.010 0.001	Small Small Small Small
	Abstract Reasoning	Area Age Sex Interaction	0.000 0.001 0.076 0.001	Small Small Medium Small
	Semantic Fluency	Area Age Sex Interaction	0.000 0.000 0.049 0.001	Small Small Small Small
Verbal memory	Immediate Recall Attempts and Delayed Recall	Area Age Sex Interaction	0.005 0.001 0.002 0.001	Small Small Small Small
	Recognition	Area Age Sex Interaction	0.002 0.073 0.000 0.003	Small Medium Small Small
Visual Memory	Immediate Recall and Delayed Recall	Area Age Sex Interaction	0.010 0.001 0.003 0.000	Small Small Small Small
	Recognition	Area Age Sex Interaction	0.005 0.005 0.026 0.004	Small Small Small Small
	Alternate Visuomotor	Area Age Sex Interaction	0.004 0.010 0.084 0.002	Small Small Medium Small

Area = refugees vs. non-refugees.

## Data Availability

The data that support the findings of this study are available from the corresponding author, upon reasonable request.

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
