# Peer review of "Differences in Neuropsychological Performance between Refugee and Non-Refugee Children in Palestine"

_ijerph, 2021, doi:10.3390/ijerph18115750_

Round 1

Reviewer 1 Report

Thank you for the opportunity to review the paper, "Differences in neuropsychological performance between refugee and non-refugee children in Palestine". I commend the authors on taking an understudied topic, yet one of vital importance and also one that is very timely. The article is generally well written and easy to follow, with a clear rationale for the nature of the study (i.e., the lack of study on neuropsychological versus psychological functioning in refugee children). The methodology seems generally sound, and I appreciate the use of a psychometric battery that was designed for and normed on Arab children, enhancing cultural validity. 

Overall I do believe that the article has merit and is worthy of publication. However, before being accepted, some points should be addressed.

METHODS:
* In lines 121-123, I'm unsure if they mean that there is a scarcity of standardized neurocognitive data on children these age, or a literal scarcity of appropriate psychometric measures. If the latter, then this makes less sense, because why would you assess cognition at the age when fewest tools are available? It will be helpful to clarify this point. 
* Are any data available on the family composition of the children? E.g., were they being raised by both of their parents? Had they lost either or both of their parents due to the conflict? The work of Dr. Bruce Perry and others indicates that social support - and more specifically - secure attachment figures - mitigates the impact of traumatic exposure for children. This is also seen in the Adverse Childhood Experiences (ACE) literature, where the loss of one or more parents constitutes an ACE. It would be interesting to know if these data are available. If they are, then it may be worthwhile to look at whether the integrity of the family system moderates the degree of cognitive impairment in refugee children, with those from intact families perhaps suffering less cognitive impairment. If such data are not available, then recommend addressing this point as a possible limitation in the Discussion.
* Is there IQ data available on the children? If so, this would be a useful addition and if not, should be addressed as a limitation in the Discussion.

RESULTS:
* The results could benefit from a major re-write to bring more clarity. For example, I am not sure when the authors are presenting a main effect (e.g., of age) versus an interaction (e.g., age x refugee status or age x sex). The tables should be more clearly marked whether a main effect or interaction is being reported; same for the narrative of the text. Also, in the main body of the text, consider for each cognitive domain reporting the results in a consistent manner (e.g., main effects, followed by interactions). This will greatly improve readability. If unsure of formatting, I suggest reviewing APA Style Guide formatting for suggestions.
* Related, I imagine that many people (like myself) outside of the Arab world will be unfamiliar with this test battery. Are the scores reported standard scores or raw scores? If this is a standardized test battery with norms, then standard scores will be more meaningful to report than raw differences, at least in terms of determining whether there is a clinically significant difference and not just a statistically significant one. The meaning of the scores should be clarified.
* On lines 192-194: "The Continuous Performance Tasks (CPT) results showed that the group of refugee children was more successful than children outside the camp. But non-refugee children made fewer mistakes than refugee children."  In this case, what does "more successful" mean - do you mean faster reaction time? Meaning that, in the speed-accuracy tradeoff, the refugee children were faster whereas the non-refugee children were more accurate?

DISCUSSION:
* An additional limitation of the study to consider is the fact that the study is cross-sectional and not longitudinal. Future studies may benefit from following a cohort of children over time to see how trauma affects their ongoing neurocognitive development.

Author Response

We are grateful to the reviewer for their insights and recommendations, which have all been taken into consideration in our revision of the paper, as detailed in our point-by-point responses.

Thank you for the opportunity to review the paper, "Differences in neuropsychological performance between refugee and non-refugee children in Palestine". I commend the authors on taking an understudied topic, yet one of vital importance and also one that is very timely. The article is generally well written and easy to follow, with a clear rationale for the nature of the study (i.e., the lack of study on neuropsychological versus psychological functioning in refugee children). The methodology seems generally sound, and I appreciate the use of a psychometric battery that was designed for and normed on Arab children, enhancing cultural validity. 

Overall I do believe that the article has merit and is worthy of publication. However, before being accepted, some points should be addressed.

Comment

METHODS:

* In lines 121-123, I'm unsure if they mean that there is a scarcity of standardized neurocognitive data on children these age, or a literal scarcity of appropriate psychometric measures. If the latter, then this makes less sense, because why would you assess cognition at the age when fewest tools are available? It will be helpful to clarify this point. 

Answer

Our ultimate goal is to evaluate children of all educational cycles, but we have started including these ages because we think that neuropsychological evaluation at these ages facilitates prevention. The sooner difficulties are identified in neuropsychological domains, the sooner preventive work can be done and improve academic performance and school adaptation. A phrase is included in the introduction that tries to argue this idea. 

“These age groups were selected because they correspond to the first years of schooling in the Palestinian education system and it is at these ages that the neuropsychological problems relating to school performance can appear. In addition, several studies have stressed the importance of conducting neuropsychological studies with schoolchildren and the need to create normative data on school children.”

Comment

* Are any data available on the family composition of the children? E.g., were they being raised by both of their parents? Had they lost either or both of their parents due to the conflict? The work of Dr. Bruce Perry and others indicates that social support - and more specifically - secure attachment figures - mitigates the impact of traumatic exposure for children. This is also seen in the Adverse Childhood Experiences (ACE) literature, where the loss of one or more parents constitutes an ACE. It would be interesting to know if these data are available. If they are, then it may be worthwhile to look at whether the integrity of the family system moderates the degree of cognitive impairment in refugee children, with those from intact families perhaps suffering less cognitive impairment. If such data are not available, then recommend addressing this point as a possible limitation in the Discussion.

Answer

Thanks for the suggestion. We have included a table with general information on the children and some sociodemographic data on the parents. 

Table 2 includes general information on the sample, as well as sociodemographic data on the parents.

Table 2. General information on the sample and sociodemographic data of the parents.

Refugee

M (S.D) or n (%)

No refugee

M (S.D) or n (%)

p

Number of people they live with

5.77 (1.91)

5.57 (1.91)

.295

Number sisters and brothers

3.9 (2.02)

4 (2.2)

.297

Hours of sleep

9.49 (1.27)

10.48 (1.94)

.000

Chronic health problem

Yes

No

137 (29.6%)

326 (70.4%)

18 (15%)

102 (85%)

.001

Father's educational level

No studies

Primary

High school

Graduate University

Magister/ Doctorate

Do not know

30 (6.5%)

114 (24.6%)

197 (42.6%)

33 (7.1%)

70 (15.1%)

20 (4.1%)

5 (4.2%)

59 (49.2%)

43 (35.8%)

4 (3.3%)

8 (6.7%)

1 (0.8%)

.000

Mother’s educational level

No studies

Primary

High school

Graduate University

Magister / Doctorate

Do not know

12 (2.6%)

86 (18.5%)

205 (44.2%)

25 (5.4%)

124 (26.7%)

12 (2.6%)

4 (3.3%)

20 (16.7%)

59 (49.2%)

7 (5.8%)

29 (24.2%)

1 (0.8%)

.778

Employeed fatther

Yes

No

402 (86.7%)

62 (13.3%)

114 (95%)

6 (5%)

.043

Father’s job

Autonomous

Family business

Unskilled manual labor

Skilled manual labor

Domestic / caregiver

Professional

Student

35 (8.7%)

2 (0.5%)

183 (45.5%)

37 (9.2%)

0

100 (24.9%)

45 (11.2%)

14 (12.3)

2 (1.7%)

70 (61.7%)

13 (11.4%)

0

10 (8.7%)

5 (4.2%)

.000

Employeed motther

Si

No

78 (16.8%)

386 (83.2%)

10 (8.3%)

110 (91.7%)

.021

Mother’s  job

Autonomous

Family business

Unskilled manual labor

Skilled manual labor

Domestic / caregiver

Professional

Student”

7 (9%)

0

5 (6.4%)

3 (3.8%)

3 (3.8%)

27 (34.6%)

33 (42.3%)

0

0

0

0

4 (40%)

6 (60%)

.808

Comment

* Is there IQ data available on the children? If so, this would be a useful addition and if not, should be addressed as a limitation in the Discussion.

Answer

We could not collect any information regarding the IQ of children. Following reviewer’s suggestion we have include it in the limitation section of the discussion.

“Third, we could not assess the intellectual coefficient of the children that participated in the research.”

Comment

RESULTS:

* The results could benefit from a major re-write to bring more clarity. For example, I am not sure when the authors are presenting a main effect (e.g., of age) versus an interaction (e.g., age x refugee status or age x sex). The tables should be more clearly marked whether a main effect or interaction is being reported; same for the narrative of the text. Also, in the main body of the text, consider for each cognitive domain reporting the results in a consistent manner (e.g., main effects, followed by interactions). This will greatly improve readability. If unsure of formatting, I suggest reviewing APA Style Guide formatting for suggestions.

* Related, I imagine that many people (like myself) outside of the Arab world will be unfamiliar with this test battery. Are the scores reported standard scores or raw scores? If this is a standardized test battery with norms, then standard scores will be more meaningful to report than raw differences, at least in terms of determining whether there is a clinically significant difference and not just a statistically significant one. The meaning of the scores should be clarified.

* On lines 192-194: "The Continuous Performance Tasks (CPT) results showed that the group of refugee children was more successful than children outside the camp. But non-refugee children made fewer mistakes than refugee children."  In this case, what does "more successful" mean - do you mean faster reaction time? Meaning that, in the speed-accuracy tradeoff, the refugee children were faster whereas the non-refugee children were more accurate?

Answer

We sincerely appreciate these suggestions from the reviewer. We have rewritten part of the results (see in the manuscript), and we have added new information and a table with the effect sizes and their interpretation. 

“In addition, our results showed that effect size was large in sustained attention (Area), while medium effect sizes were reported in Phonemic fluency (sex), visuomotor coordination (sex), Alternate Visuomotor (sex), Working Memory (sex), abstract reasoning (sex), and verbal memory recognition (age). Further, results revealed a small effect size in the rest of the neuropsychological tests (see Table 5).

Table 5. Effect size and interpretation

Interpretation

eta squared  )

Variable

Test

Domain

Small

Small

Medium

Small

.004

.044

.069

.001

Area

Age

Sex

Interaction

Phonemic Fluency

Language

Small

Small

Small

Small

.002

.001

.051

.005

Area

Age

Sex

Interaction

Verbal Comprehension – Figures

Small

Small

Small

Small

.016

.001

.032

.000

Area

Age

Sex

Interaction

Verbal Comprehension Images

Large

Small

Small

Small

.144

.003

.002

.003

Area

Age

Sex

Interaction

Continuous performance Test

(Correct answers)

Sustained

Attention

Small

Small

Small

Small

.052

.000

.012

.002

Area

Age

Sex

Interaction

Continuous performance Test

(Omission)

Large

Small

Small

Small

.195

.013

.000

.001

Area

Age

Sex

Interaction

Continuous performance Test

(Reaction Time)

Small

Small

Medium

Small

.003

.004

.088

.000

Area

Age

Sex

Interaction

visuomotor coordination

Visuomotor

Small

Small

Medium

Small

.002

.003

.087

.001

Area

Age

Sex

Interaction

Working

Memory

ExecutiveFunctions

Small

Small

Small

Small

.006

.003

.010

.001

Area

Age

Sex

Interaction

Planning

Small

Small

Medium

Small

.000

.001

.076

.001

Area

Age

Sex

Interaction

Abstract

Reasoning

Small

Small

Small

Small

.000

.000

.049

.001

Area

Age

Sex

Interaction

SemanticFluency

Small

Small

Small

Small

.005

.001

.002

.001

Area

Age

Sex

Interaction

Immediate recall attempts & delayed recall

Verbal memory

Small

Medium

Small

Small

.002

.073

.000

.003

Area

Age

Sex

Interaction

Recognition

Small

Small

Small

Small

.010

.001

.003

.000

Area

Age

Sex

Interaction

Immediate recall & delayed Recall

Visual Memory

Small

Small

Small

Small

.005

.005

.026

.004

Area

Age

Sex

Interaction

Recognition

Small

Small

Medium

Small

.004

.010

.084

.002

Area

Age

Sex

Interaction

Alternate Visuomotor

Area = refugees vs. no refugees”

Comment

DISCUSSION:

* An additional limitation of the study to consider is the fact that the study is cross-sectional and not longitudinal. Future studies may benefit from following a cohort of children over time to see how trauma affects their ongoing neurocognitive development.

Answer

We agree with the reviewer’s suggestion and we have included this limitation in the discussion section.

“Finally, the cross-sectional research design doesn’t allow us to study the direct influence of age in the neuropsychological development of these populations. Studies using a longitudinal design are needed to test the effects of trauma in the cognitive development of refugees and non-refugees children in Palestine.”

Reviewer 2 Report

I would like to thank the Editor and the Authors for the opportunity to review the manuscript entitled: “Differences in neuropsychological performance between refugee and non-refugee children in Palestine”.

I find the contribution a relevant contribution for the Journal, nonetheless, I think that the manuscript needs come revisions in order to make the implications of the study even more effective.

Authors should briefly describe the background of the refugee and non-refugee children. It is well known that many psychosocial variables may impact the neurobiology of children. Knowing about the context would help those readers who may not be familiar with the specificity of the investigated population.

In line, are there any data on the parents of the investigated children?

It would important to know if parents signed the informed consent.

Please, report and discuss effect sizes.

Finally, I think that the manuscript would benefit if the Authors would dedicate more space to the implications of their study.

Author Response

We are grateful to the reviewer for their insights and recommendations, which have all been taken into consideration in our revision of the paper, as detailed in our point-by-point responses.

I would like to thank the Editor and the Authors for the opportunity to review the manuscript entitled: “Differences in neuropsychological performance between refugee and non-refugee children in Palestine”.

I find the contribution a relevant contribution for the Journal, nonetheless, I think that the manuscript needs come revisions in order to make the implications of the study even more effective.

Comment

- Authors should briefly describe the background of the refugee and non-refugee children. It is well known that many psychosocial variables may impact the neurobiology of children. Knowing about the context would help those readers who may not be familiar with the specificity of the investigated population.

In line, are there any data on the parents of the investigated children?

Answer

We appreciate this recommendation. In this sense, we have included a table in which we present additional information on both groups and sociodemographic data on the parents.

Table 2 includes general information on the sample, as well as sociodemographic data on the parents.

Table 2. General information on the sample and sociodemographic data of the parents.

Refugee

M (S.D) or n (%)

No refugee

M (S.D) or n (%)

p

Number of people they live with

5.77 (1.91)

5.57 (1.91)

.295

Number sisters and brothers

3.9 (2.02)

4 (2.2)

.297

Hours of sleep

9.49 (1.27)

10.48 (1.94)

.000

Chronic health problem

Yes

Not

137 (29.6%)

326 (70.4%)

18 (15%)

102 (85%)

.001

Father's educational level

No studies

Primary

high school

University Graduate

Magister / Doctorate

Do not know

30 (6.5%)

114 (24.6%)

197 (42.6%)

33 (7.1%)

70 (15.1%)

20 (4.1%)

5 (4.2%)

59 (49.2%)

43 (35.8%)

4 (3.3%)

8 (6.7%)

1 (0.8%)

.000

Mother’s educational level

No studies

Primary

high school

University Graduate

Magister / Doctorate

Do not know

12 (2.6%)

86 (18.5%)

205 (44.2%)

25 (5.4%)

124 (26.7%)

12 (2.6%)

4 (3.3%)

20 (16.7%)

59 (49.2%)

7 (5.8%)

29 (24.2%)

1 (0.8%)

.778

Employeed fatther

Si

No

402 (86.7%)

62 (13.3%)

114 (95%)

6 (5%)

.043

Father’s  job

Autonomous

Family business

Unskilled manual labor

Skilled manual labor

Domestic / caregiver

Professional

Student

35 (8.7%)

2 (0.5%)

183 (45.5%)

37 (9.2%)

0

100 (24.9%)

45 (11.2%)

14 (12.3)

2 (1.7%)

70 (61.7%)

13 (11.4%)

0

10 (8.7%)

5 (4.2%)

.000

Employeed motther

Yes

No

78 (16.8%)

386 (83.2%)

10 (8.3%)

110 (91.7%)

.021

Mother’s  job

Autonomous

Family business

Unskilled manual labor

Skilled manual labor

Domestic / caregiver

Professional

Student”

7 (9%)

0

5 (6.4%)

3 (3.8%)

3 (3.8%)

27 (34.6%)

33 (42.3%)

0

0

0

0

4 (40%)

6 (60%)

.808

Comment

It would important to know if parents signed the informed consent.

Answer

The study was approved by the Institutional Review Board of The University of Bethlehem.  The children's parents signed an informed consent.This information is collected in the manuscript. 

Institutional Review Board Statement: “The study was conducted according to the guidelines of the Declaration of Helsinki, and approved by the Institutional Review Board of The University of Bethlehem).”

Informed Consent Statement: “Informed consent was obtained from all subjects involved in the study.”

Please, report and discuss effect sizes.

Answer

We have added new information and a table with the effect sizes and their interpretation. 

“In addition, our results showed that effect size was large in sustained attention (Area), while medium effect sizes were reported in Phonemic fluency (sex), visuomotor coordination (sex), Alternate Visuomotor (sex), Working Memory (sex), abstract reasoning (sex), and verbal memory recognition (age). Further, results revealed a small effect size in the rest of the neuropsychological tests (see Table 5).

Table 5. Effect size and interpretation

Interpretation

eta squared  )

Variable

Test

Domain

Small

Small

Medium

Small

.004

.044

.069

.001

Area

Age

Sex

Interaction

Phonemic Fluency

Language

Small

Small

Small

Small

.002

.001

.051

.005

Area

Age

Sex

Interaction

Verbal Comprehension – Figures

Small

Small

Small

Small

.016

.001

.032

.000

Area

Age

Sex

Interaction

Verbal Comprehension Images

Large

Small

Small

Small

.144

.003

.002

.003

Area

Age

Sex

Interaction

Continuous performance Test

(Correct answers)

Sustained

Attention

Small

Small

Small

Small

.052

.000

.012

.002

Area

Age

Sex

Interaction

Continuous performance Test

(Omission)

Large

Small

Small

Small

.195

.013

.000

.001

Area

Age

Sex

Interaction

Continuous performance Test

(Reaction Time)

Small

Small

Medium

Small

.003

.004

.088

.000

Area

Age

Sex

Interaction

visuomotor coordination

Visuomotor

Small

Small

Medium

Small

.002

.003

.087

.001

Area

Age

Sex

Interaction

Working

Memory

ExecutiveFunctions

Small

Small

Small

Small

.006

.003

.010

.001

Area

Age

Sex

Interaction

Planning

Small

Small

Medium

Small

.000

.001

.076

.001

Area

Age

Sex

Interaction

Abstract

Reasoning

Small

Small

Small

Small

.000

.000

.049

.001

Area

Age

Sex

Interaction

SemanticFluency

Small

Small

Small

Small

.005

.001

.002

.001

Area

Age

Sex

Interaction

Immediate recall attempts & delayed recall

Verbal memory

Small

Medium

Small

Small

.002

.073

.000

.003

Area

Age

Sex

Interaction

Recognition

Small

Small

Small

Small

.010

.001

.003

.000

Area

Age

Sex

Interaction

Immediate recall & delayed Recall

Visual Memory

Small

Small

Small

Small

.005

.005

.026

.004

Area

Age

Sex

Interaction

Recognition

Small

Small

Medium

Small

.004

.010

.084

.002

Area

Age

Sex

Interaction

Alternate Visuomotor

Area = refugees vs. no refugees”

Comment

Finally, I think that the manuscript would benefit if the Authors would dedicate more space to the implications of their study.

Answer

Following reviewer’s suggestion we have include a paragraph at the end of the discussion section to address the health-related implications of the present results

“The main risk factors for normal neurocognitive development are related to conditions of the prenatal stage, perinatal circumstances, nutritional, infectious and toxic factors, and child-rearing practices [42], as well as socioeconomic conditions, in especially those related to poverty, in general, and extreme poverty in particular [43] and with exposure to violence [23]. Many of these circumstances are present in the sample of this study. This work constitutes the first neuropsychological study carried out with children living in Palestinian refugee camps. It has made it possible to obtain a baseline that can be used for future longitudinal studies. The results confirm the need to generate programs to intervene in the detected deficits and implement other preventive programs from an earlier age. These results have important health implications, indicating the vulnerability and the negative effects in the neurodevelopment in these children. Given the large number of variables and factors that interact (poverty, lack of space, poor infrastructures and education services) it is necessary to develop interventions for health promotion that can address both internal and external risks factors.”  

Reviewer 3 Report

This manuscript investigates neuropsychological performance in vulnerable children (i.e., refugee children). Neuropsychological performance seem that was measured with language (verbal comprehension – figures [phonemic fluency, comfig???, and verbal comprehension imagines], sustained attention (continuous performance test, correct answers, and omission), visuomotor (visuomotor coordination and alternate visuomotor), memory – recognition (verbal memory and visual memory), and executive functions (working memory, planning, abstract reasoning, and semantic fluency) and vulnerability with two conditions (refugee vs. non-refugee). Sample were 584 Palestinian school children, 464 (???%) refugees and 120 non-refugees, 295 (???%) men and 289 women, from 7 to 8 years old (M =???, SD = ???). Main design was a between 2 x 2 x 3 factorial in which the independent variables were vulnerability (refugee vs. non-refugee), sex (male vs. female) and age (six, seven and eight) and dependent variables were the different indicators of neuropsychological performance. As was to be expected, vulnerable children (i.e., Palestinian children living in refugee camps) present lower neuropsychological performance than other children (i.e., the other Palestinian children).

This interesting manuscript analyzes a case of highly vulnerable children (i.e., refugee children) in a high at-risk area (Palestine). it could be titled something like: “The most vulnerable children in a high-risk area: The refugee children neuropsychological performance in Palestine. This work could form an important part of the literature on vulnerable (e.g., Fackler et al., 2021; Garcia et al., 2020; Uceda-Maza et al., 2017) children at risk areas (e.g., Gracia et al., 1995; Lorence et al., 2019; Riquelme et al., 2018).

My biggest concern is the general lack of paper variables organization along the manuscript. Measures of neuropsychological performance are not in the same order during text and table 2. Please, always the same order during all text. Anymore, parts of main variables are different labeled in different parts or manuscript. It has been very difficult for me to prepare a short summary. Current presentation is a puzzle in present form.

When you used an extra within factor, in measures section, you could indicate that was added immediate recall trials 1, 2, 3, delayed trials. This could be labeled, for example, as three extra indicators of verbal memory performance that also you have examined for more detailed.

From a more general theoretical framework the deterioration of these children would be better understood and present manuscript would have a greater impact on the specialized literature on general concept of health (e.g., Fackler et al., 2021; Garcia et al., 2020).

References

Gracia, E., García, F., & Musitu, G. (1995). Macrosocial determinants of social integration: Social class and area effect. Journal of Community and Applied Social Psychology, 5, 105-119.   doi:10.1002/casp.2450050204

Fackler, C. A., Baugh, N., Lovegren, A. A., Nemeroff, C., & Whatley Blum, J. (2021). Technology-Enhanced Health Promotion for College Students: A Seed Development Project. Nursing Reports, 11(1), 143–151. MDPI AG. Retrieved from http://dx.doi.org/10.3390/nursrep11010014

Lorence, B., Hidalgo, V., Pérez-Padilla, J., & Menéndez, S. (2019). The role of parenting styles on behavior problem profiles of adolescents. International Journal of Environmental Research and Public Health, 16(2767), 1-17.   doi:10.3390/ijerph16152767

Riquelme, M., García, O. F., & Serra, E. (2018). Psychosocial maladjustment in adolescence: Parental socialization, self-esteem, and substance use. Anales de Psicología, 34, 536-544.   doi:10.6018/analesps.34.3.315201

Uceda-Maza, F. X., & Alonso, J. D. (2017). The link between vulnerability and social exclusion and criminal trajectories. An association study. Psychosocial Intervention, 26, 29-37. doi:10.1016/j.psi.2016.07.002

Garcia, O. F., Serra, E., Zacares, J. J., Calafat, A., & Garcia, F. (2020). Alcohol use and abuse and motivations for drinking and non-drinking among Spanish adolescents: Do we know enough when we know parenting style? Psychology and Health, 35, 645-664.   doi:10.1080/08870446.2019.1675660

Author Response

We are grateful to the reviewer for their insights and recommendations, which have all been taken into consideration in our revision of the paper, as detailed in our point-by-point responses.

Comment

This manuscript investigates neuropsychological performance in vulnerable children (i.e., refugee children). Neuropsychological performance seem that was measured with language (verbal comprehension – figures [phonemic fluency, comfig???, and verbal comprehension imagines], sustained attention (continuous performance test, correct answers, and omission), visuomotor (visuomotor coordination and alternate visuomotor), memory – recognition (verbal memory and visual memory), and executive functions (working memory, planning, abstract reasoning, and semantic fluency) and vulnerability with two conditions (refugee vs. non-refugee). Sample were 584 Palestinian school children, 464 (???%) refugees and 120 non-refugees, 295 (???%) men and 289 women, from 7 to 8 years old (M =???, SD = ???). Main design was a between 2 x 2 x 3 factorial in which the independent variables were vulnerability (refugee vs. non-refugee), sex (male vs. female) and age (six, seven and eight) and dependent variables were the different indicators of neuropsychological performance. As was to be expected, vulnerable children (i.e., Palestinian children living in refugee camps) present lower neuropsychological performance than other children (i.e., the other Palestinian children).

Answer

We appreciate the suggestion to the reviewer. We have included new information in the manuscript.

“A total of 584 Palestinian schoolchildren (295 boys – 50.51%- and 289 girls -49.48%-) aged 6, 7 and 8 voluntarily participated in the study (M=6.99, SD=0.817). As noted above, these ages were selected because they correspond to the first years of schooling in the Palestinian education system and due to the scarcity of standard scale tests for these ages. Participants were selected from two Palestinian provinces (Hebron and Bethlehem).These two geographical areas include five refugee camps, in Bethlehem: the Dheisheh Camp (established in 1949), the Beit Jibrin Camp (established in 1950), the Aida Camp (established in 1950), and in Hebron: the Arroub Camp (established in 1949), and the Fawwar Camp (established in 1949). A total of 464 (79.45%) children were selected from schools located within five refugee camps. In addition, 120 (20.55%) children living outside refugee camps participated in the study. Table 1 presents the geographical distribution of the study sample.”

Comment

- This interesting manuscript analyzes a case of highly vulnerable children (i.e., refugee children) in a high at-risk area (Palestine). it could be titled something like: “The most vulnerable children in a high-risk area: The refugee children neuropsychological performance in Palestine. This work could form an important part of the literature on vulnerable (e.g., Fackler et al., 2021; Garcia et al., 2020; Uceda-Maza et al., 2017) children at risk areas (e.g., Gracia et al., 1995; Lorence et al., 2019; Riquelme et al., 2018).

 AnswerWe have included new references that help contextualize the work. “The main risk factors for normal neurocognitive development are related to conditions of the prenatal stage, perinatal circumstances, nutritional, infectious and toxic factors, and child-rearing practices [42], as well as socioeconomic conditions, in especially those related to poverty, in general, and extreme poverty in particular [43] and with exposure to violence [23].”

Comment

- My biggest concern is the general lack of paper variables organization along the manuscript. Measures of neuropsychological performance are not in the same order during text and table 2. Please, always the same order during all text. Anymore, parts of main variables are different labeled in different parts or manuscript. It has been very difficult for me to prepare a short summary. Current presentation is a puzzle in present form.

When you used an extra within factor, in measures section, you could indicate that was added immediate recall trials 1, 2, 3, delayed trials. This could be labeled, for example, as three extra indicators of verbal memory performance that also you have examined for more detailed.

Answer

We sincerely appreciate these suggestions from the reviewer. We have rewritten part of instruments and the results and we have added new information and a table with the effect sizes and their interpretation (see in the manuscript).  

Comment

- From a more general theoretical framework the deterioration of these children would be better understood and present manuscript would have a greater impact on the specialized literature on general concept of health (e.g., Fackler et al., 2021; Garcia et al., 2020).

Answer

Following reviewer’s suggestion we have include a paragraph at the end of the discussion section to address the health-related implications of the present results

The main risk factors for normal neurocognitive development are related to conditions of the prenatal stage, perinatal circumstances, nutritional, infectious and toxic factors, and child-rearing practices [42], as well as socioeconomic conditions, in especially those related to poverty, in general, and extreme poverty in particular [43] and with exposure to violence [23]. Many of these circumstances are present in the sample of this study. This work constitutes the first neuropsychological study carried out with children living in Palestinian refugee camps. It has made it possible to obtain a baseline that can be used for future longitudinal studies. The results confirm the need to generate programs to intervene in the detected deficits and implement other preventive programs from an earlier age. These results have important health implications, indicating the vulnerability and the negative effects in the neurodevelopment in these children. Given the large number of variables and factors that interact (poverty, lack of space, poor infrastructures and education services) it is necessary to develop interventions for health promotion that can address both internal and external risks factors.  

Round 2

Reviewer 2 Report

I want to thank the Authors to have scrupulously revised their work, that has efficaciously improved. 

Reviewer 3 Report

The authors have satisfactorily addressed my concerns.